# Fusion Gene-Based Classification of Variant Cytogenetic Rearrangements in Acute Myeloid Leukemia

**DOI:** 10.3390/genes14020396

**Published:** 2023-02-03

**Authors:** Mary Gudipati, Melody Butler, Rima Koka, Maria R. Baer, Yi Ning

**Affiliations:** 1Department of Pathology, University of Maryland School of Medicine, Baltimore, MD 21201, USA; 2Department of Medicine, University of Maryland School of Medicine, Greenebaum Comprehensive Cancer Center, Baltimore, MD 21201, USA

**Keywords:** *RUNX1*::*RUNX1T1* fusion, cryptic translocation, complex rearrangement, acute myeloid leukemia

## Abstract

Acute myeloid leukemia (AML) represents a heterogeneous disease entity that is continuously moving to a more genetically defined classification. The classification of AML with recurrent chromosomal translocations, including those involving core binding factor subunits, plays a critical role in diagnosis, prognosis, treatment stratification, and residual disease evaluation. Accurate classification of variant cytogenetic rearrangements in AML contributes to effective clinical management. We report here the identification of four variant t(8;V;21) translocations in newly diagnosed AML patients. Two patients showed a t(8;14) and a t(8;10) variation, respectively, with a morphologically normal-appearing chromosome 21 in each initial karyotype. Subsequent fluorescence in situ hybridization (FISH) on metaphase cells revealed cryptic three-way translocations t(8;14;21) and t(8;10;21). Each resulted in *RUNX1*::*RUNX1T1* fusion. The other two patients showed karyotypically visible three-way translocations t(8;16;21) and t(8;20;21), respectively. Each resulted in *RUNX1*::*RUNX1T1* fusion. Our findings demonstrate the importance of recognizing variant forms of t(8;21) translocations and emphasize the value of applying *RUNX1*::*RUNX1T1* FISH for the detection of cryptic and complex rearrangements when abnormalities involving chromosome band 8q22 are observed in patients with AML.

## 1. Introduction

Acute myeloid leukemia (AML) is a clonal hematopoietic disorder resulting from genetic alterations in normal hematopoietic stem cells. These alterations disrupt normal differentiation and/or cause excessive proliferation of abnormal immature leukemic cells. The classification of AML has shifted from a morphology-based classification to a classification algorithm based primarily on genetic abnormalities. The current World Health Organization (WHO) classification includes a major entity termed AML with recurrent genetic abnormalities, meaning with specific chromosomal or mutational alterations [1,2,3].

Translocation t(8;21)(q22;q22) is a recurrent cytogenetic abnormality and is one of the most common subtypes of AML, occurring in approximately 5% of AML patients. This subtype has predominantly neutrophilic maturation and is associated with a high rate of complete remission and favorable long-term outcomes [1,2,3,4,5,6]. The t(8;21) translocation leads to the formation of an oncogenic fusion of *RUNX1* (runt-related transcription factor 1) on 21q22 to *RUNX1T1* (runt-related transcription factor 1; translocated to 1) on 8q22. The fusion resides on the derivative 8q22. It leads to the disruption of the normal function of the core-binding factor in hematopoietic differentiation and maturation and recruits transcriptional repressors to block the expression of genes involved in normal hematopoiesis, thus impairing apoptosis [6].

At the molecular level, the translocation breakpoint in *RUNX1* occurs between exons 5 and 6; the translocation juxtaposes the 59 end of the *RUNX1* gene, including the RHD domain, with the 39 end of the *RUNX1T1* gene with its four NHR domains. The generated fusion protein consists of 752 amino acids; the first 177 amino acids are derived from RUNX1, whereas 575 amino acids are from RUNX1T1. Structurally, RUNX1::RUNX1T1 therefore has five domains: the RHD from RUNX1 and NHR domains 1 to 4 from RUNX1T1 [6]

While the t(8;21) translocation is a primary chromosomal abnormality in a specific subtype of AML, secondary, non-random cytogenetic abnormalities have also been identified. Frequent secondary abnormalities include the loss of a sex chromosome (-X or -Y) or a deletion in 9q [7].

Variant and complex rearrangements in the form of t(8;V;21), with V being a variable third chromosome, have been reported in approximately 3–4% of AML patients with t(8;21) [7,8]. Similarly to t(8;21), these variant rearrangements result in *RUNX1*::*RUNX1T1* fusion on the derivative 8q22. Identification and accurate classification of these *RUNX1*::*RUNX1T1* fusion-positive variants in the t(8;21) subtype plays an important role in the diagnosis of this specific subtype of AML and in the optimal clinical management of patients with this subtype.

We report here the identification of four variant t(8;V;21) translocations in newly diagnosed AML patients. Significantly, two patients showed t(8;14) and t(8;10) respectively, with a morphologically normal-appearing chromosome 21 in each initial karyotype. Subsequent fluorescence in situ hybridization (FISH) on metaphase cells using a *RUNX1*-*RUNX1T1* probe revealed cryptic three-way translocations t(8;14;21) and t(8;10;21), which resulted in *RUNX1*::*RUNX1T1* fusion in both patients. The other two patients showed karyotypically visible three-way translocations t(8;16;21) and t(8;20;21). Each resulted in *RUNX1*::*RUNX1T1* fusion. We conclude that *RUNX1*::*RUNX1T1* FISH is critical in the detection and classification of variant t(8;V;21).

## 2. Materials and Methods

Bone marrow or peripheral blood samples from the four patients described in this report were received in our cytogenetics laboratory for diagnostic purposes. The study was conducted according to the guidelines of the Declaration of Helsinki and approved by the Institutional Review Board of the University of Maryland, Baltimore (HP104494).

Each received sample was set into cultures and cultured in RPMI 1640 medium for 24 to 48 h. The cultured cells were harvested and G-banded following standard cytogenetic procedures as described in the AGT Cytogenetics Laboratory Manual [9]. Metaphase cells were analyzed and karyograms were prepared. FISH assays were performed using *RUNX1T1*-*RUNX1* dual-color dual-fusion probes (Cytocell, Oxford, UK) on metaphase and interphase cells from each sample following the standard procedures described in the AGT Cytogenetics Laboratory Manual [10]. Two hundred interphase nuclei and available metaphase cells were examined and representative FISH images were documented.

NGS-based cDNA sequencing was performed in the molecular lab. RNA was extracted from patients’ bone marrow or blood samples, which was followed by a reverse transcription reaction to generate cDNA. The library was prepared on the Ion Chef System. Massive parallel cDNA sequencing was performed using an Ion S5 OMA Workflow (Thermo Fisher Scientific, Waltham, MA, USA) for the detection of a set of 29 fusion genes, including *RUNX1*::*RUNX1T1* fusion.

## 3. Results

### 3.1. Patient 1

A peripheral blood sample was received from a 25-year-old man admitted to our Cancer Center in February 2022. His hemoglobin level was 8.5 g/dL, the platelet count was 51 K/uL, and the white blood cell count (WBC) was 10.8 K/uL with 42% blasts. The blasts expressed the myeloid markers CD33, CD117, and cytoplasmic MPO along with the immature marker CD34 by flow cytometry. Additionally, there was aberrant dim and partial expression of CD19, typically a B-cell marker. These findings indicated a diagnosis of AML.

Initial karyotype analysis showed an apparent t(8;14) translocation, with a sub-clone showing t(8;14) and -Y. Both copies of chromosome 21 appeared normal (Figure 1a). Because the translocation breakpoint was at band 8q22 and loss of chromosome Y is a frequent secondary finding associated with t(8;21), FISH was subsequently performed using a *RUNX1*-*RUNX1T1* probe to rule out a variant rearrangement. FISH analysis of metaphase cells revealed a cryptic three-way translocation (Figure 1b), and the karyotype was therefore revised to 46,XY,t(8;14;21)(q22;q32;q22) [1]/45,X,-Y,t((8;14;21). This three-way translocation resulted in a *RUNX1*-*RUNX1T1* fusion in 76% of the 200 examined nuclei.

Molecular testing confirmed the presence of gene fusion *RUNX1*::*RUNX1T1 RUNX1*::*RUNX1T1*.R3R3 chr21:36231771—chr8:93029591.

### 3.2. Patient 2

A bone marrow sample was received from a 57-year-old man admitted to our Cancer Center in February 2022. He had hypercellular bone marrow with 70% cellularity. His hemoglobin level was 5.2 g/dL, the platelet count was 14 K/uL, and the WBC was 2.8 K/uL with 31% blasts. The blasts expressed the myeloid markers CD33, CD117, and cytoplasmic MPO along with the immature marker CD34 by flow cytometry. Additionally, there was aberrant CD56 expression, typically a natural killer cell marker, along with dim expression of CD19. These findings indicated a diagnosis of AML.

Initial karyotype analysis showed an apparent t(8;10) translocation, with a sub-clone showing t(8;10) and -Y. Both copies of chromosome 21 appeared normal (Figure 2a). Because the translocation breakpoint was at band 8q22 and loss of chromosome Y is a frequent secondary finding associated with t(8;21), FISH was subsequently performed using a *RUNX1*-*RUNX1T1* probe to rule out a variant rearrangement. FISH analysis of metaphase cells revealed a cryptic three-way translocation (Figure 2b), and the karyotype was therefore revised to 46,XY,t(8;10;21)(q22;q25;q22) [10]/45,X,-Y,t(8;10;21) [8]/46,XY [2]. This three-way translocation resulted in a *RUNX1*-*RUNX1T1* fusion in 96% of the 200 examined nuclei.

Molecular testing confirmed the presence of gene fusion *RUNX1*::*RUNX1T1 RUNX1*::*RUNX1T1*.R3R3 chr21:36231771—chr8:93029591.

### 3.3. Patient 3

A bone marrow sample was received from a 49-year-old woman admitted to our Cancer Center in May 2022. She had hypercellular bone marrow with 80% cellularity. Her hemoglobin level was 12.6 g/dL, the platelet count was 21 K/uL, and the WBC was 2.8 K/uL with 18% blasts. The blasts expressed the myeloid markers CD33, CD117, and cytoplasmic MPO along with the immature marker CD34 by flow cytometry. Additionally, there was aberrant CD56 expression along with dim, partial expression of CD19.

Karyotype analysis detected a three-way translocation: 46,XX,t(8;16;21)(q22;q12-13;q22) [11] (Figure 3). FISH analysis showed a *RUNX1*-*RUNX1T1* fusion in 76% of the 200 examined nuclei.

Because no sample was submitted for this patient, molecular testing was not possible.

### 3.4. Patient 4

A peripheral blood sample was received from a 20-year-old man admitted to our Cancer Center in December 2022. His hemoglobin level was 6.8 g/dL, the platelet count was 10 K/uL, and the WBC was 5.1 K/uL with 25% blasts. The blasts expressed the myeloid markers CD33, CD117, and cytoplasmic MPO along with partial expression of the immature marker CD34 by flow cytometry. Additionally, there was aberrant CD56 expression along with partial expression of CD19.

Karyotype analysis detected a three-way translocation: 45,X,-Y,t(8;20;21)(q22;q13.2;q22) [12] (Figure 4). FISH analysis showed a *RUNX1*-*RUNX1T1* fusion in 75% of the 200 examined nuclei.

Molecular testing confirmed the presence of gene fusion *RUNX1*::*RUNX1T1 RUNX1*::*RUNX1T1*.R3R3 chr21:36231771—chr8:93029591.

## 4. Discussion

The t(8;21)(q22;q22) translocation, with resulting *RUNX1T1*::*RUNX1* fusion, is a recurrent cytogenetic abnormality in AML. Although the *RUNX1T1*::*RUNX1* fusion is most commonly seen in cases with cytogenetically visible t(8;21), detection of the fusion transcript in the absence of visible t(8:21) has been reported in a few cases with cryptic rearrangements, including sequence insertions [13] and three-way translocations [12,14,15,16]. Most reported variant three-way t(8;V;21) translocations have been described with V being a variable third chromosome. Interestingly, the cryptic t(8;14;21) translocation detected in Patient 1 appeared to be similar to, if not the same as, a patient described by Lau et al. [15], indicating that this three-way translocation might represent a rare but recurrent cytogenetic rearrangement.

Accurate classification of AML involving core-binding factor (CBF) subunits, especially those with cryptic and complex translocations, is of important significance in clinical management. AML with *RUNX1T1*::*RUNX1* fusion is known to be associated with a complete remission rate and a high cure rate after high-dose cytarabine consolidation therapy; thus, allogeneic hematopoietic stem cell transplantation in first remission is not recommended for these patients. Due to a limited number of reported cases and studies, it is not clear whether the clinical outcomes of patients with three-way translocations are different from those of patients with the t(8;21) translocation. At the molecular level, all t(8;V;21) translocations resulted in a *RUNX1T1*::*RUNX1* fusion, which is the same as the consequence of t(8;21) translocations. All of our four patients achieved remission. While three of the four patients currently remain in remission, Patient 2 passed away after achieving remission due to end-stage renal disease status post failed renal transplantation. In comparison to the typical t(8;21) translocations, the long-term clinical outcomes of patients with three-way translocations remains to be evaluated.

AML with t(8;21) that disrupts core-binding factor (CBF) subunit α and inv(16) that disrupts CBF subunit bata are collectively referred to as CBF AML. When compared to other cytogenetic groups, patients with CBF AML have relatively favorable outcomes. Multiple studies have shown that the presence of *KIT* gene mutations in CBF AML confers a higher relapse risk [17,18]. Therefore, screening for *KIT* mutations in CBF AML has both prognostic and therapeutic significance. The activated *KIT* can potentially be targeted with novel tyrosine kinase inhibitors. The molecular testing results from our three tested patients with *RUNX1::RUNXT1* fusion showed no evidence of *KIT* mutation in these patients.

The four patients with t(8;V;21) translocations were identified among approximately one hundred and twenty AML patients that were treated in our Cancer Center in 2022. Our findings indicate that the presence of t(8;21) translocation variants may be more frequent than previously appreciated. We recommend *RUNX1*::*RUNX1T1* FISH when abnormalities involving 8q22 are observed in AML karyotypes. We also recommend taking immunophenotype findings into consideration. It is worth noting that all four patients with t(8;V;21) translocations showed aberrant CD19 expression by flow cytometric analysis, and three of the four patients (Patients 2, 3, and 4) also showed aberrant CD56 expression. CD19 is normally a B-cell lineage marker, and CD56 is a natural killer cell/stem cell marker. Aberrant expression of CD19 on myeloblasts or co-expression of CD19 and CD56, along with myeloid antigens, has been identified as a characteristic immunophenotype associated with t(8;21) [11,19,20]. Therefore, *RUNX1*::*RUNX1T1* FISH is also recommended for AML patients with aberrant CD19 expression or those with CD19 and CD56 co-expression. When the classic t(8;21) translocation is not apparent in a karyotype, immunophenotype detection of these markers may be helpful in suggesting that FISH be performed to look for a variant or cryptic *RUNX1*::*RUNX1T1* fusion.

In summary, we identified four variant cytogenetic rearrangements, namely t(8;14;21), t(8;10;21), t(8;16;21), and t(8;20;21), in four patients with newly diagnosed AML. Each rearrangement resulted in a *RUNX1*::*RUNX1T1* fusion. The first two rearrangements were cryptic three-way translocations that were not readily identifiable through conventional karyotyping. These findings indicate the challenges associated with accurate classification, which is required to support the stratified treatment of AML. We recommend an integral approach that combines cytogenetic analysis with molecular and immunophenotypic studies to improve the detection of variant cytogenetic rearrangements in AML.

## Figures and Tables

**Figure 1 genes-14-00396-f001:**
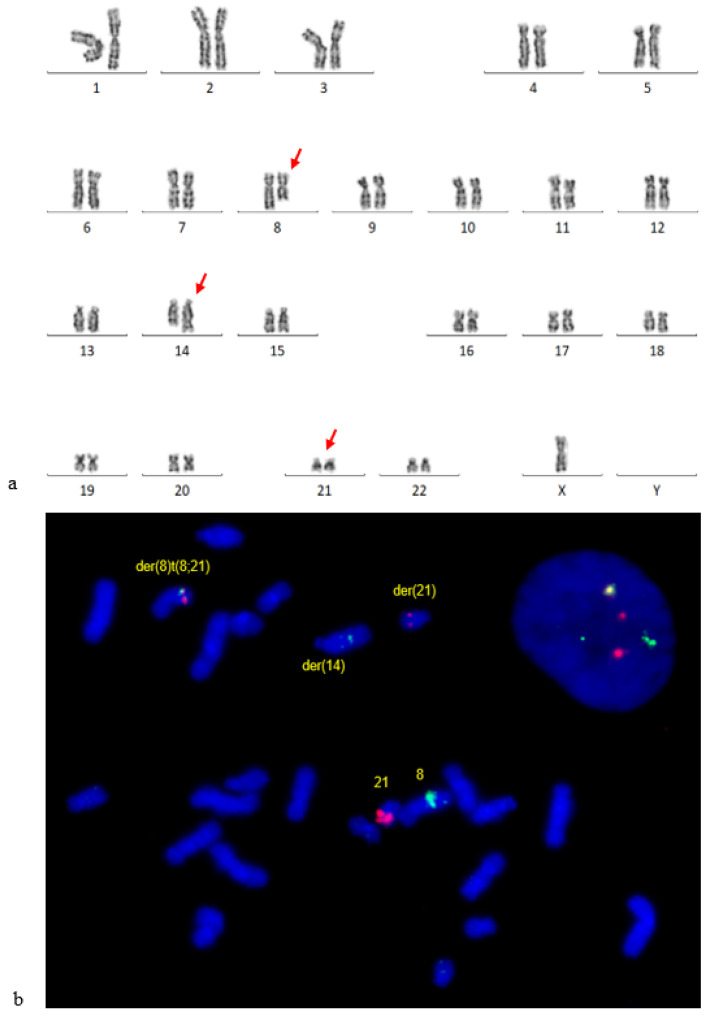
Cytogenetic findings of Patient 1. (**a**) A representative karyogram: 45,X,-Y,t(8;14;21)(q22;q32;q22). The derivative 21q was cryptic. (**b**) FISH using a dual-fusion probe *RUNX1*(red)/*RUNX1T1*(green) revealed *RUNX1::RUNX1T1* fusion on the derivative 8q.

**Figure 2 genes-14-00396-f002:**
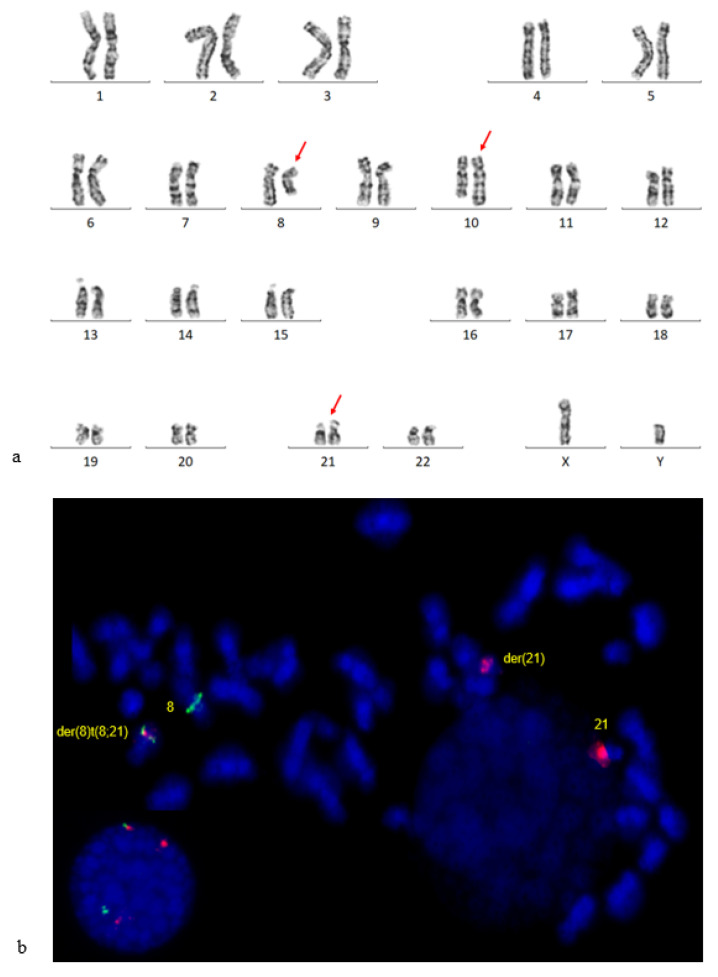
Cytogenetic findings of Patient 2. (**a**) A representative karyogram: 45,XY,t(8;10;21)(q22;q25;q22). The derivative 21q was cryptic. (**b**) FISH using a dual-fusion probe *RUNX1*(red)/*RUNX1T1*(green) revealed *RUNX1::RUNX1T1* fusion on the derivative 8q.

**Figure 3 genes-14-00396-f003:**
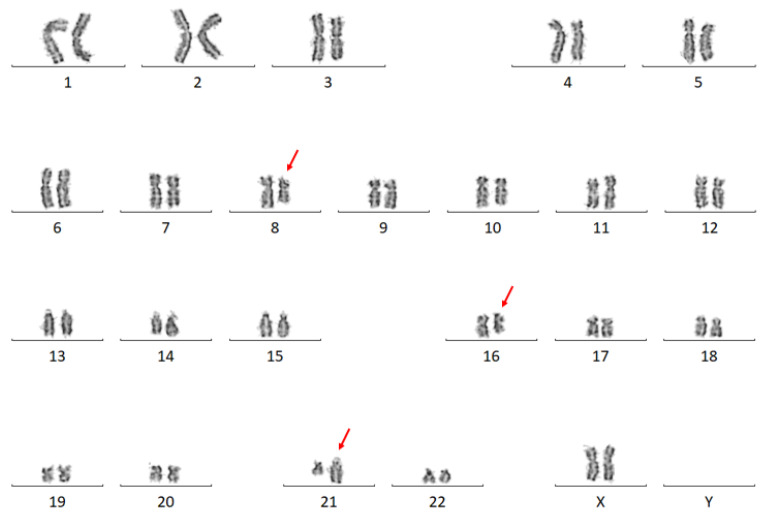
A representative karyogram of Patient 3: 46,XX,t(8;16;21)(q22;q12-13;q22).

**Figure 4 genes-14-00396-f004:**
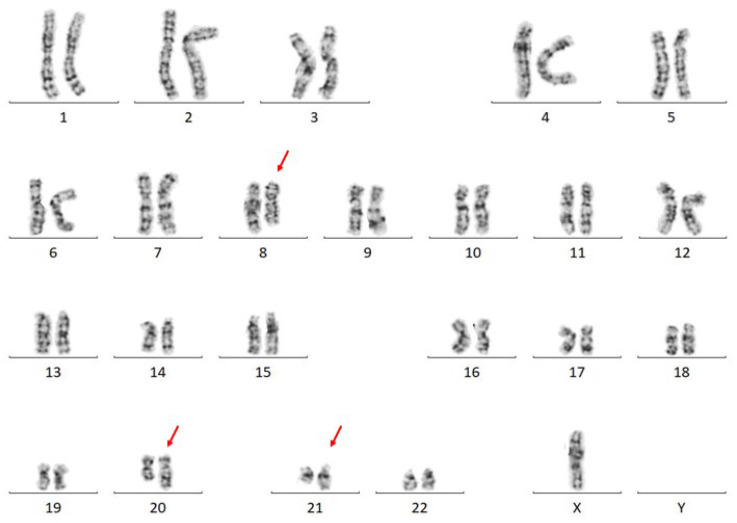
A representative karyogram of Patient 4: 46,X,-Y,t(8;20;21)(q22;q13.2;q22).

## Data Availability

There were no additional data.

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
