# Peer review of "Fusion Gene-Based Classification of Variant Cytogenetic Rearrangements in Acute Myeloid Leukemia"

_genes, 2023, doi:10.3390/genes14020396_

Round 1

Reviewer 1 Report

Well written, nicely presented short case series. I have two major and three minor comments

Major comments: 

1. Please present metaphase FISH images for cases 3 and 4. 

2. RT-PCR experiments are recommended for RUNX1::RUNX1T1 fusion to show the presence of a viable fusion transcript

Minor comments:

1.Please attach IRB approval, at lease approval number, "for not-human subject research" determination from University of Maryland 

2. FISH with Whole chromosome painting (WCP) probes would be beneficial  to show three way translocations unequivocally. (not mandatory as WCP probes are fairly expensive and presented metaphase chromosome images are highly suggestive of variant t(8;v;21) translocations) 

3. Please use "RUNX1::RUNX1T1" fusion constantly in the text. I saw RUNX1T1::RUNXT1 in lines 110-111 and RUNX1-RUNX1T1 fusion in lines 73, 84, 91, 98

Author Response

Major comments: 

  1. Please present metaphase FISH images for cases 3 and 4. 

FISH was performed on direct, uncultured samples (with no culture to obtain metaphases) for all four patients. Metaphase FISH images were documented for Patients 1 and 2, because both had cryptic translocations that were not detectable by conventional karyotyping. Since conventional karyotyping showed visible three-way translocations for Patients 3 and 4, and interphase FISH confirmed RUNX1::RUNX1T1 fusion in both cases, metaphase FISH was not performed for Patients 3 and 4.

  1. RT-PCR experiments are recommended for RUNX1::RUNX1T1 fusion to show the presence of a viable fusion transcript

Currently, our cytogenetics lab performs karyotyping and FISH for clinical samples, and our molecular lab performs NGS-based testing to detect mutations and fusion genes for the same samples with indication of AML. Molecular testing was performed on Patients 1, 2, and 4, and the RUNX1::RUNX1T1 fusion was confirmed in these three patients by cDNA sequencing. However, the molecular lab did not receive a sample for Patient 3.

The findings have been incorporated into the manuscript under “Results” for each patient.

Minor comments:

  1. Please attach IRB approval, at least approval number, "for not-human subject research" determination from University of Maryland 

Please see revised part in manuscript and the attached IRB letter.

  1. FISH with Whole chromosome painting (WCP) probes would be beneficial to show three way translocations unequivocally. (not mandatory as WCP probes are fairly expensive and presented metaphase chromosome images are highly suggestive of variant t(8;v;21) translocations) 

We currently use gene-specific probe to help clinical management. We no longer have WCP probes in our lab.

  1. Please use "RUNX1::RUNX1T1" fusion constantly in the text. I saw RUNX1T1::RUNXT1 in lines 110-111 and RUNX1-RUNX1T1 fusion in lines 73, 84, 91, 98

Corrected.  Thank you.

Reviewer 2 Report

The work brings important data on rare gene fusions that may play a fundamental role in the clinical outcome and prognosis of patients with AML, however the work leaves many points for improvement in the presentation of results, in their discussion and in the conclusion of the outcome of these alterations in the study patients.

Major Points

Summary needs to be improved in the presentation mainly of the results and conclusion of these.

I suggest that the introduction be better structured based on the literature on the study of the disease, there are many points that can be made in the introduction and the biggest problem is the frequent presence of unquoted paragraphs.

In the "material and methods" section the authors need to cite the protocol followed for the karyotyping and FISH procedures that were performed. In addition, I suggest that the opinion of the ethics committee and the Declaration of Helsinki be included, as patient data are being used, even with the justification of the diagnostic criteria already performed by the center, ethical issues are relevant to this point.

In the results section: I suggest that the description of each patient is followed immediately by the karyotype and FISH figures (I suggest improving the quality of the FISH figures in all patients).

The results are not discussed properly, what is the impact of these genetic findings on the 4 patients in this study? What prognosis is associated with these? Was there an unfavorable outcome compared to other patients at the center? The work needs to improve these points to be a relevant communication to the scientific community.

Finally, the conclusion of the findings point to what? What is the importance of this study for the community or for the clinical follow-up of patients

Author Response

Major Points

Summary needs to be improved in the presentation mainly of the results and conclusion of these.

Revised (see all revised text in red).

I suggest that the introduction be better structured based on the literature on the study of the disease, there are many points that can be made in the introduction and the biggest problem is the frequent presence of unquoted paragraphs.

Revised with added references.

In the "material and methods" section the authors need to cite the protocol followed for the karyotyping and FISH procedures that were performed. In addition, I suggest that the opinion of the ethics committee and the Declaration of Helsinki be included, as patient data are being used, even with the justification of the diagnostic criteria already performed by the center, ethical issues are relevant to this point.

Revised as suggested and attached IRB approval letter.

In the results section: I suggest that the description of each patient is followed immediately by the karyotype and FISH figures (I suggest improving the quality of the FISH figures in all patients).

Revised as suggested.

The results are not discussed properly, what is the impact of these genetic findings on the 4 patients in this study? What prognosis is associated with these? Was there an unfavorable outcome compared to other patients at the center? The work needs to improve these points to be a relevant communication to the scientific community.

See revised text in red.

Finally, the conclusion of the findings point to what? What is the importance of this study for the community or for the clinical follow-up of patients

See revised text in red.

Round 2

Reviewer 1 Report

Authors addressed my concerns.

Author Response

Reviewer: Authors addressed my concerns.

Thank you!

Reviewer 2 Report

The authors added the methodology of NGS-based cDNA sequencing, where a panel of 29 fusions were investigated, however the results of these tests confirmed in only 3 patients the presence of the RUNX1::RUNX1T1 fusion, no other fusions were identified?

Why not explore the NGS results to be discussed as well? 

Author Response

The authors added the methodology of NGS-based cDNA sequencing, where a panel of 29 fusions were investigated, however the results of these tests confirmed in only 3 patients the presence of the RUNX1::RUNX1T1 fusion, no other fusions were identified?  Why not explore the NGS results to be discussed as well? 

Our NGS panel for AML includes 40 genes and 29 fusion genes. RUNX1::RUNX1T1 was the only fusion gene detected in the tested samples. No other clinically relevant variants were detected. 

We included the molecular findings, because the other reviewer requested molecular confirmation of the RUNX1::RUNXT1 fusion.  We responded “Currently, our cytogenetics lab performs karyotyping and FISH for clinical samples, and our molecular lab performs NGS-based testing to detect mutations and fusion genes for the same samples with indication of AML. Molecular testing was performed on Patients 1, 2, and 4, and the RUNX1::RUNX1T1 fusion was confirmed in these three patients by cDNA sequencing. The molecular lab did not receive a sample for Patient 3”.   

We included these findings in the results, because they are relevant to the focus of this manuscript to raise the awareness of variant t(8;21) in cytogenetic studies to avoid misclassification.   

It is well-known that t(8;21) with RUNX1::RUNX1T1 fusion is a driver for this subtype of AML. Mutations in KIT gene can alter the good prognosis of t(8;21) with RUNX1::RUNX1T1 fusion. No KIT mutation was detected in any of our four patients.  

Reference:  Adverse prognostic significance of KIT mutations in adult acute myeloid leukemia with inv(16) and t(8;21): a Cancer and Leukemia Group B Study. J Clin Oncol. 2006 Aug 20;24(24):3904-11.